# Variations in Dysbindin-1 are associated with cognitive response to antipsychotic drug treatment

Diego Scheggia[1,12], Rosa Mastrogiacomo[1], Maddalena Mereu[1,2], Sara Sannino[1], Richard E. Straub[3], Marco Armando[4], Francesca Managò[1], Simone Guadagna[1], Fabrizio Piras[5], Fengyu Zhang[3], Joel E. Kleinman[3], Thomas M. Hyde[3], Sanne S. Kaalund[6], Maria Pontillo[4], Genny Orso[7], Carlo Caltagirone[5], Emiliana Borrelli[8], Maria A. De Luca[9], Stefano Vicari[4], Daniel R. Weinberger[3,10], Gianfranco Spalletta[5,11] & Francesco Papaleo[1]

Antipsychotics are the most widely used medications for the treatment of schizophrenia spectrum disorders. While such drugs generally ameliorate positive symptoms, clinical responses are highly variable in terms of negative symptoms and cognitive impairments. However, predictors of individual responses have been elusive. Here, we report a pharmacogenetic interaction related to a core cognitive dysfunction in patients with schizophrenia. We show that genetic variations reducing dysbindin-1 expression can identify individuals whose executive functions respond better to antipsychotic drugs, both in humans and in mice. Multilevel ex vivo and in vivo analyses in postmortem human brains and genetically modified mice demonstrate that such interaction between antipsychotics and dysbindin-1 is mediated by an imbalance between the short and long isoforms of dopamine D2 receptors, leading to enhanced presynaptic D2 function within the prefrontal cortex. These findings reveal one of the pharmacodynamic mechanisms underlying individual cognitive response to treatment in patients with schizophrenia, suggesting a potential approach for improving the use of antipsychotic drugs.

[1] Department of Neuroscience and Brain Technologies, Genetics of Cognition laboratory, Istituto Italiano di Tecnologia, via Morego, 30, 16163 Genova, Italy. [2] Dipartimento di Scienze del Farmaco, Universita' degli Studi di Padova, Largo Meneghetti 2, 35131 Padova, Italy. [3] Lieber Institute for Brain Development, Johns Hopkins University Medical Campus, Baltimore, MD 21205, USA. [4] Department of Neuroscience, Bambino Gesù Children's Hospital, Piazza Sant'Onofrio 4, 00100 Rome, Italy. [5] IRCCS Santa Lucia Foundation, Neuropsychiatry Laboratory, 00179 Rome, Italy. [6] Research Laboratory for Stereology and Neuroscience, Bispebjerg University Hospital, 2400 Copenhagen, NV, Denmark. [7] IRCCS E. Medea Scientific Institute, 23842 Bosisio Parini, Italy. [8] University of California, Irvine, CA 92697, USA. [9] Department of Biomedical Sciences, Università di Cagliari, 09124 Cagliari, Italy. [10] Departments of Psychiatry, Neurology, Neuroscience and the McKusick-Nathans Institute of Genetic Medicine, Johns Hopkins School of Medicine, Baltimore, MD 21205, USA. [11] Menninger Department of Psychiatry and Behavioral Sciences, Baylor College of Medicine, Houston, TX 77030, USA. [12] Present address: Center for Psychiatric Neuroscience, Department of Psychiatry, University Hospital Center Lausanne, Prilly-Lausanne CH-1008, Switzerland. Correspondence and requests for materials should be addressed to F.P. (email: francesco.papaleo@iit.it)

Antipsychotics are the first-line and most widely used medications for the management of schizophrenia spectrum and other psychotic disorders[1]. In agreement with the heterogeneous nature of these disorders, clinical responses to antipsychotics drugs are highly variable[2]. Thus, clinical guidelines strongly recommend adapting antipsychotic treatments to each individual case[1]. However, to date, only very scarce biomarkers exist to implement more effective and personalized healthcare.

Efforts have been made to improve the assessment and definition of treatment-resistant schizophrenia. However, the identification of criteria for schizophrenia cognitive deficits is still incomplete, in part because of the lack of cognitive symptom domains in the most widely used clinical rating scales (e.g., the PANSS, BPRS, SANS, and SAPS)[3]. Nonetheless, the consensus is that clinical responses for negative symptoms and cognitive impairments are suboptimal and highly variable. Notably, cognitive deficits are considered one of the main sources of disability, having the most critical impact on public health and long-term outcomes[4,5]. Treatments with first- and second-generation antipsychotics produce small neurocognitive improvements in both chronic and first-episode schizophrenia patients[6–10]. Nevertheless, cognitive responses to antipsychotic drugs show marked interindividual variability[11,12], and the mechanistic basis of this seemingly unpredictable variability is unknown.

All antipsychotic drugs interact with dopamine D2 receptors[13], with variable ranges of D2 occupancy suggested to be important for optimal clinical and cognitive responses[13–15]. From a pharmacokinetic perspective, genetic variations such as those in CYP2D6, CYP3A4/5, and ABCB1 might impact the metabolism and distribution of antipsychotic drugs, potentially affecting the margin between the dosages that are required for efficacy and those associated with side effects[16,17]. Shifting to a pharmacodynamic perspective, genetic variations influencing D2 receptors could in principle also influence the efficacy of antipsychotic drugs[13]. Genetic variations in the dystrobrevin binding protein 1 (DTNBP1) gene, encoding dysbindin-1[18–20] (Dys), a synaptic protein regulating synaptic vesicles and receptors recycling[21,22], can alter D2 receptor availability. Moreover, genetic variations in Dys might interfere with the dopaminergic system, altering cognitive functions in both mice and healthy humans[19,20,23–25]. Dystrophin and its binding partners (such as Dys) have been implicated in schizophrenia by recent copy number variant (CNV) studies[26], by older linkage studies[27–32], and by postmortem human brain expression data[33,34]. However, GWAS have challenged this association[35]. Other fields such as cancer genomics and cardiovascular medicine have suggested that genetic variants predicting drug responses might be unrelated to the cause of the target pathology[36]. Based on the physiological impact of Dys on D2 receptors, we hypothesize in this work that variation in the Dys gene may be related to variation in cognitive response to antipsychotic drugs.

Through a translational mouse-human approach, we studied the effects of common functional genetic variations resulting in reduced Dys expression on cognitive responses to antipsychotic drugs. We show that genetic variations associated with reduced Dys expression in the human brain differentiate individuals whose executive functions respond better to these drugs. Mechanistically, using lentiviral-vector-mediated microRNA (miR) silencing and neurochemical strategies for region-specific investigation in mice and postmortem human brain gene expression analyses, we demonstrate that this interaction between antipsychotics and Dys relies selectively on the function of dopamine D2 receptors within the prefrontal cortex (PFC). In particular, Dys genetic reduction is associated with an antipsychotic-dependent increase in the ratio between the D2Short (D2S) and D2Long (D2L) isoforms in the PFC of both humans and mice, which results in a potentiation of cortical presynaptic D2 signaling. A genetic approach in mice demonstrates that the D2S/D2L imbalance in favor of D2S is the cause of the cognitive improvements that occur in response to antipsychotic administration in the context of reduced Dys. Certainly, Dys is not expected to be the only relevant molecular probe involved in responses to antipsychotic drugs. Nevertheless, our findings provide another step toward personalized treatment[37,38] for schizophrenia, suggesting Dys-related mechanisms as a tool for a more focused approach to alleviating cognitive disabilities based on a defined biological mechanism that moderates the response to antipsychotics treatment.

## Results

**Dys-antipsychotics interaction in human executive functions.** We first investigated whether functional Dys genetic variants would differentiate cognitive abilities in patients with schizophrenia undergoing chronic antipsychotic drug treatment. We used a three-marker (rs2619538-rs3213207-rs1047631) haplotype at the DTNBP1 gene locus (Dys Hap) previously associated with a pattern of cognitive-related PFC functional activation consistent with reduced Dys in mouse models[19]. RNA sequencing from the dorsolateral PFC (DLPFC) of 594 human subjects demonstrated that carriers of the Dys Hap (T-A-A), hereinafter referred to as Dys Hap+/− and −/− (demographic details in Supplementary Table 1 and genotype frequencies in Supplementary Table 2), had reduced Dys expression (Fig. 1a) compared to Dys Hap+/+ individuals. We recruited 259 patients receiving chronic treatment with antipsychotic medication, and we assessed their executive function abilities, as these are cognitive hallmarks of schizophrenia[39]. Notably, executive function deficits in patients with schizophrenia are particularly evident in extradimensional set shifting (EDS) and in the analogous category shift of the Wisconsin Card Sorting Task (WCST)[40,41]. We found that Dys Hap+/− and −/− patients, with putatively reduced expression of Dys, performed better than Dys Hap+/+ patients (Fig. 1b–d). In particular, Dys Hap−/− patients made fewer perseverative (Fig. 1b) and non-perseverative errors (Fig. 1c) and completed more WCST categories (Fig. 1d) than Dys Hap+/+ patients did. Similarly, Dys Hap+/− patients made fewer non-perseverative errors than Dys Hap+/+ patients did. No Dys Hap-dependent differences were present in demographic characteristics, PANSS scores, or the duration or dosage of current antipsychotic treatments (Supplementary Table 1). All SNPs were in Hardy–Weinberg equilibrium (HWE; Supplementary Table 2).

To check for replicability in an independent cohort of subjects, we next investigated this association using the neurocognitive battery used in the Clinical Antipsychotic Trials of Intervention Effectiveness (CATIE), a double-blind multicenter trial sponsored by the National Institute of Mental Health (NIMH) to assess responses to antipsychotic drugs. Each patient was randomly assigned to treatment with a single antipsychotic drug, and any previous therapy was discontinued during the first 2 weeks of double-blind treatment. After a baseline assessment, patients were followed for up to 18 months. We used a sample of 359 patients who completed the study and for whom genetic data were available. We found that Dys Hap−/− and+/− patients, characterized by reduced Dys expression, displayed a better WCST performance compared to Dys Hap+/+ patients (Fig. 1e and Supplementary Table 3). Thus, the CATIE study confirmed that genetic variations associated with reduced Dys are associated with better executive function response to antipsychotic treatment in patients with schizophrenia.

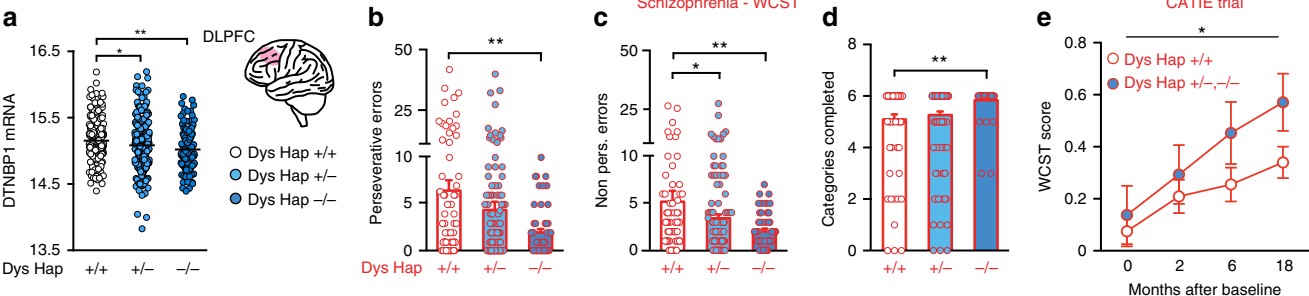

**Fig. 1** Genetic variations reducing dysbindin-1 are associated with better executive functions in antipsychotic-treated subjects. **a** Reduction of dysbindin-1 expression in the DLPFC of Dys Hap+/− and −/− human subjects (light and dark blue circles, respectively) compared to Dys Hap+/+ (white circles) ($n = 594$; one-way ANOVA, $F_{(2, 591)} = 5.70$, $p < 0.005$). Analysis with the R package haplo.stats v1.7.7 yielded a global $p$ value of $3.5 \times 10^{-7}$ for DTNBP1 at the gene level, which was in part driven by the specific Dys haplotype ($p = 2.54 \times 10^{-5}$), associated with lower expression (i.e., Dys Hap+/+ > Dys Hap +/− > Dys Hap−/−). **b–d** Patients with schizophrenia with reduced dysbindin-1 expression (Dys Hap−/−) made **b** fewer perseverative ($n = 259$; one-way ANOVA, $F_{(2, 251)} = 4.64$, $p < 0.01$), and **c** fewer non-perseverative errors (one-way ANOVA, $F_{(2, 251)} = 4.21$, $p < 0.01$) on the WCST, and **d** passed more categories (one-way ANOVA, $F_{(2, 251)} = 4.74$, $p < 0.01$). No Dys-haplotype-dependent differences were present in demographical characteristics or in the duration or dosage of antipsychotic treatments (see Supplementary Table 1). All single-nucleotide polymorphisms (SNPs) were in Hardy–Weinberg equilibrium (HWE) for healthy subjects and patients with schizophrenia (see Supplementary Table 2). **e** Patients with schizophrenia with reduced dysbindin-1 (Dys Hap+/− and −/−, blue circles), who were enrolled in the NIH CATIE trial showed better WCST performance than those with the Dys Hap+/+ (white circles) genotype. The WCST score was calculated by averaging z-scores for preservative errors and categories achieved ($n = 359$; two-way ANOVA, time effect, $F_{(3, 1046)} = 5.03$, $p < 0.005$; genotype effect, $F_{(1, 1046)} = 4.38$, $p < 0.04$). Error bars represent S.E.M. *$p < 0.05$, **$p < 0.005$

**Antipsychotics improves cognition based on Dys genetics**. To reduce the possibility that multidrug treatments and the long duration of the illness could have confounded the results, we recruited another independent cohort of patients with schizophrenia in their first episode of psychosis. These patients were naive to medication when recruited and were treated for the first time with one antipsychotic (risperidone or aripiprazole) for 4 weeks, then evaluated with the WCST. Similar to chronic patients, first-episode patients with the Dys Hap−/− genotype had better attentional set-shifting abilities than non-carriers (Dys Hap+/+) (Fig. 2a–c).

Ideally, it would have been interesting to have an untreated or placebo group of patients with schizophrenia never treated with antipsychotics. However, to recruit a sufficient number of these subjects for further genetic stratification would be challenging and would involve a number of ethical issues. Thus, we used mice to test the possibility that genetic variations selectively reducing Dys expression would confer executive function benefits following administration of antipsychotics. We tested mice with reduced levels of dysbindin-1 (Dys+/−) and control littermates (Dys+/+) using the Intra-/Extra-Dimensional Attentional Set-Shifting task (ID/ED *Operon* task)[42], which is equivalent to the human WCST. Starting from 2 weeks before the test, mice were treated with risperidone, a commonly used antipsychotic, or with vehicle. Chronic treatment with risperidone not only rescued the attentional set-shifting impairment of Dys+/− mice but also improved their EDS performance compared to that of vehicle- and risperidone-treated Dys+/+ mice (Fig. 2d–f and Supplementary Fig. 1a–c). Risperidone treatment had no effect on Dys +/+ mice (Fig. 2d–f and Supplementary Fig. 1a–c). We further tested whether any residual effect of injections and manipulations might have existed. Non-manipulated mice with genetically reduced Dys showed a selective deficit in EDS abilities compared to control wild-type littermates (Fig. 3a–c and Supplementary Fig. 2a–c). Furthermore, as predicted by these mouse data, healthy Dys Hap−/− humans (demographic characteristics in Supplementary Table 1) showed a cognitive disadvantage in attentional set shifting on the WCST compared to matched healthy Dys Hap+/+ subjects (Fig. 3d–f). These results support the conclusion that genetic variations reducing Dys may improve

the responsiveness of executive function to treatments with antipsychotic drugs.

**Cortical D2 are necessary for Dys-antipsychotics effects**. The mechanistic basis of the unpredictable variability in clinical responses to antipsychotics is still largely unexplored. Prompted by the behavioral effects, we sought to identify a mechanism for this pharmacogenetic interaction. We hypothesized that the antipsychotics-dysbindin-1 interaction might converge on dopamine D2 signaling within the PFC. Indeed, (1) D2 receptors are a common target of antipsychotic drugs[43]; (2) cortical D2 receptors, but not D1 receptors, are modulated by Dys genetic variations[18,20]; (3) down-regulation of Dys modulates sensitivity to D2-like agonists[20,44]; (4) genetic variations of Dys interfere with the dopaminergic system, leading to altered cognition in both mice and healthy humans[19]; (5) as in humans, the selective alteration in EDS performance in mice points to involvement of the mPFC[45]; (6) no genotype, treatment and genotype-by-treatment effects were evident in Dys, D1, glutamatergic, and serotoninergic receptors expression in the mPFC (Supplementary Fig. 3 and Supplementary Table 4). Nevertheless, antipsychotics target multiple receptors[43], and D1 receptor signaling has been implicated more than D2 in PFC-dependent cognitive functions[46]. Thus, to assess the selective role of D2 signaling, we inactivated D2 receptors in Dys+/− mice by bilaterally injecting into the mPFC a lentivirus delivering a synthetic miR coupled to a GFP tag. Two weeks later, the virus-injected mice were chronically treated with risperidone or vehicle and then tested in the ID/ED task (Fig. 4a). Silencing D2 receptors in the mPFC of Dys+/− mice was sufficient to eliminate the beneficial effect of risperidone on attentional set-shifting abilities (Fig. 4b, c and Supplementary Fig. 4). This demonstrates that the dysbindin-antipsychotic interaction depends on D2 functioning in the mPFC.

**Antipsychotic-Dys interaction potentiates presynaptic D2**. We then examined the effect of antipsychotic treatment on cortical D2 signaling in subjects with reduced levels of Dys. In mice, Dys reduction, risperidone treatment and their interaction all failed to affect the expression of D2 (Fig. 5a) in the mPFC. However,

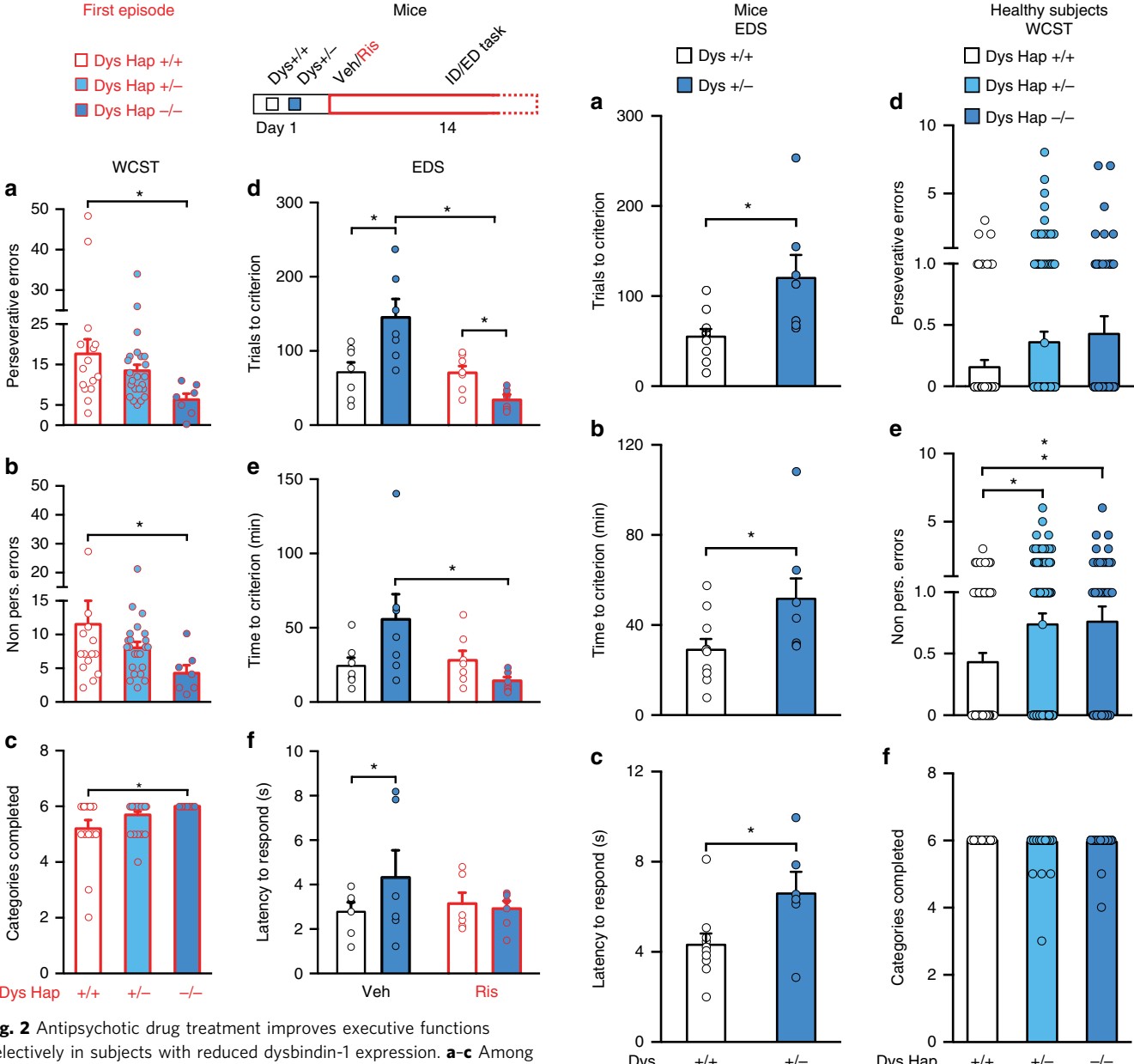

**Fig. 2** Antipsychotic drug treatment improves executive functions selectively in subjects with reduced dysbindin-1 expression. **a–c** Among patients with schizophrenia after their first episode of psychosis, those with reduced dysbindin-1 expression (Dys Hap−/−, blue dark bar) made **a** less perseverative errors (one-way ANOVA, $F_{(2, 34)} = 3.59$, $p < 0.05$; $n = 45$), **b** non-perseverative errors (one-way ANOVA, $F_{(2, 34)} = 3.59$, $p < 0.05$) and **c** passed more categories (one-way ANOVA, $F_{(2, 34)} = 3.15$, $p = 0.05$) on the WCST than those with Dys Hap+/+ (white bar). **d**, **e** Dys+/− mice (dark blue bar) after risperidone treatment (Ris in red) required **d** fewer trials (two-way ANOVA, genotype × treatment, $F_{(1, 24)} = 13.40$, $p < 0.005$, $n = 7$ per group) and **e** less time (two-way ANOVA, genotype × treatment, $F_{(1, 24)} = 5.69$, $p < 0.05$) than Dys+/+ (white bar) to complete the EDS of the ID/ED task. **f** Dys+/− mice (dark blue bar) needed more time to make a response (latency to respond) in the EDS than vehicle-treated (white bar) wild-type controls needed ($t = 2.54$, df = 15.60, $p < 0.05$). However, we did not find a treatment effect (two-way ANOVA, genotype × treatment, $F_{(1, 20)} = 1.56$, $p = 0.20$). Error bars represent S.E.M. *$p < 0.05$

**Fig. 3** Variations in dysbindin-1 modulate cognition in opposite manner in drug-naive subjects compared to patients with schizophrenia. **a**, **b** Dys+/− mice ($n = 7$, dark blue bar) needed (**a**) more trials (one-way ANOVA, genotype effect, $F_{(1, 15)} = 7.69$, $p < 0.05$) and (**b**) more time than +/+ mice (one-way ANOVA, genotype effect, $F_{(1, 15)} = 7.69$, $p < 0.05$) to complete the EDS of the ID/ED *Operon* Task ($n = 10$, white bar). **c** Increased latency to respond during the EDS in Dys+/− mice (one-way ANOVA, genotype effect, $F_{(1, 30)} = 8.43$, $p < 0.05$). **d**, **e** Dys Hap+/− and −/− healthy volunteers (light and dark blue bar, respectively) made (**d**) similar perseverative errors (one-way ANOVA, $F_{(2, 311)} = 1.64$, $p = 0.19$; $n = 314$) but (**e**), more non-perseverative errors (one-way ANOVA, $F_{(2, 311)} = 2.94$, $p = 0.05$) on the WCST than Dys Hap+/+ subjects (white bar). **f** The number of categories passed on the WCST was similar between groups (one-way ANOVA, $F_{(2, 311)} = 0.90$, $p = 0.41$). Error bars represent S.E.M.

because D2 receptors exist in two different splicing isoforms[47], we also assessed whether the dysbindin-antipsychotics interaction might differently impact the D2L and D2S isoforms. Chronic treatment with risperidone increased the D2S/D2L ratio in the mPFC of Dys+/− but not Dys+/+ mice (Fig. 5b). To investigate

whether a similar effect could be observed in the human brain, we analyzed the gene expression of D2 receptor isoforms in a sample ($n = 101$) of DLPFC tissue from patients with schizophrenia. As in mice, we found that the D2S/D2L ratio was increased in Dys Hap−/− patients if and only if antipsychotic treatments could be

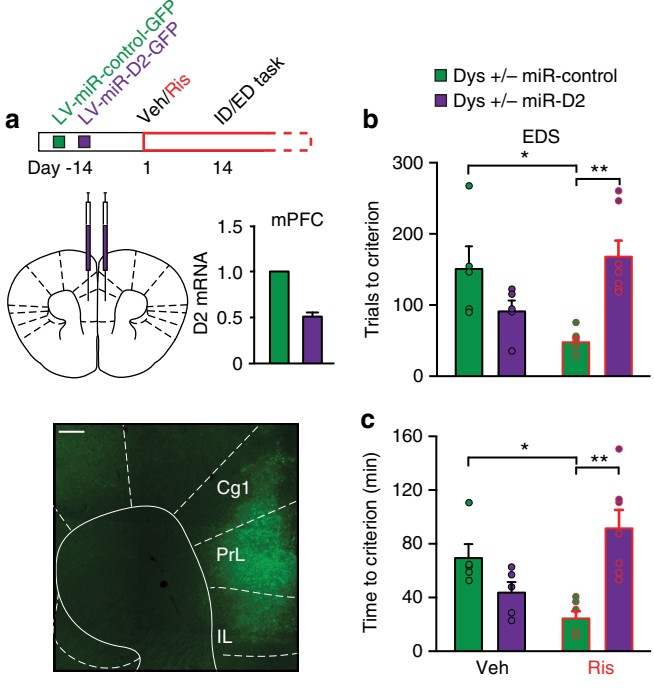

**Fig. 4** Risperidone-dysbindin-1 interaction converges on D2 receptors in the mPFC. **a** Top, Dys+/− mice have been bilaterally injected in the mPFC with a lentivirus delivering a synthetic microRNA (miR) to inactivate specifically D2 receptors (LV-miR-D2) or a control miR (LV-miR-control), coupled to a GFP tag. After 2 weeks, animals received daily injections of Veh or Ris for 14 days before the beginning of the ID/ED task and then throughout the experiment. Bottom, Injection, expression of LV-miR-D2-GFP and D2 receptors mRNA relative expression in the mPFC of Dys+/− mice. Scale bar 200 μm. Cg1, cingulate cortex; PrL, prelimbic; IL, infralimbic. **b**, **c** Dys +/− risperidone-treated mice after D2 silencing (Dys+/− miR-D2/Ris, green bar, Ris in red) needed (**b**) more trials (one-way ANOVA, $F_{(2, 15)} =$ 13,21, $p < 0.0005$, $n = 5–7$ each group) and (**c**) more time ($F_{(2, 15)} = 11,84$, $p < 0.005$) to complete the EDS of the ID/ED task than risperidone-treated Dys+/− that received a control miR (Dys+/− miR-control/Ris, purple

detected after postmortem analysis (Fig. 5c). The expression of D2 receptor isoforms was not altered by the Dys Hap in medication-naive healthy subjects ($n = 199$; Fig. 5d). Thus, we uncovered an interaction between antipsychotic drug treatments and Dys genetic variations in the PFC that alters the expression of D2 receptor isoforms in favor of D2S.

The D2S isoform is the predominant D2 presynaptic autoreceptor[47,48], providing negative feedback control of dopamine synthesis and release. Moreover, synaptic secretion of accumulated antipsychotics exerts an autoinhibitory effect on vesicular exocytosis, affecting neurotransmitter release[49]. Thus, to assess the functional and dynamic consequences of the dysbindin-by-antipsychotics interaction for the cortical D2S/D2L ratio, we performed in vivo microdialysis in the mPFC of freely moving Dys+/+ and +/− mice after chronic risperidone or vehicle treatments (Fig. 6a). Risperidone treatment in Dys+/− mice restored basal dopamine to wild-type levels while having no effect in Dys+/+ (Fig. 6b). Notably, infusion of the D2-preferring agonist quinpirole in the mPFC by reverse dialysis unraveled a larger reduction of dopamine release in Dys+/− than in +/+ following risperidone treatment (Fig. 6c), but not in vehicle-treated mice (Fig. 6d). This enhanced presynaptic D2 response in Dys+/− mice was selectively dependent on the effects of risperidone on mPFC D2 receptors, as mPFC lentiviral D2 silencing abolished it (Fig. 6e). These results provide in vivo evidence supporting the increased D2S/D2L ratio and demonstrate that risperidone treatment enhanced cortical presynaptic D2 functioning exclusively when Dys was reduced. These results add important insights into previous evidence showing that potentiation of D2 pathways in the PFC of humans, monkeys, and rodents might facilitate executive functions[50–52].

**Increased D2S/D2L ratio in Dys improves executive functions.** To demonstrate selectively that a D2S/D2L imbalance in favor of D2S is causally linked to the improved executive functions found in the context of reduced Dys, we backcrossed Dys+/− mutants with D2L+/− mice (Fig. 7a). D2L+/− mice have genetically reduced expression of the D2L isoform and upregulation of D2S[47]. In agreement with this expression pattern, we confirmed

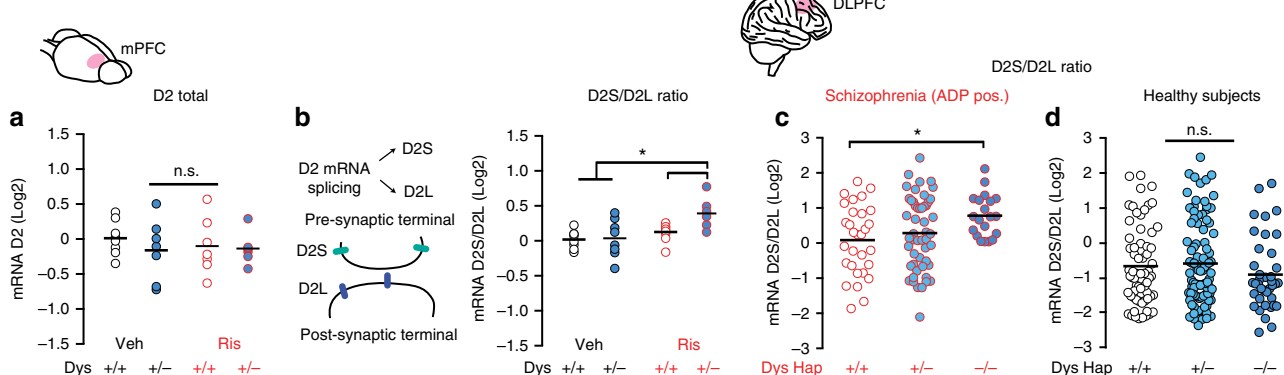

**Fig. 5** Antipsychotics generate an imbalance in cortical D2 receptor isoforms only in subjects with reduced dysbindin-1 expression. **a** Neither dysbindin-1 reduction (Dys+/− dark blue circles, Dys+/+ white circles) nor risperidone treatment (Ris in red) affected total D2 receptor expression in the mPFC ($n =$ 6–7 each group). **b** Increased D2S/D2L ratio in the mPFC of Dys+/− after chronic treatment with Ris ($n = 6–7$ each group; two-way ANOVA, treatment, $F_{(1, 25)} = 8.67$, $p < 0.05$; genotype, $F_{(1, 25)} = 4.25$, $p < 0.05$). **c** The D2S/D2L ratio in the DLPFC was increased in schizophrenia patients with reduced dysbindin-1 expression (Dys Hap −/−, dark blue circles) who tested positive in antipsychotic screening (ADP pos.) compared to Dys Hap+/+ patients (white circles; one-way ANOVA, $F_{(2, 98)} = 3.26$, $p < 0.05$; $n = 101$). We did not observe any difference in D2S/D2L ratio from postmortem human brains of the DLPFC of patients with schizophrenia in which antipsychotics where not detected (one-way ANOVA, $F_{(2, 55)} = 0.90$, $p = 0.41$; $n = 58$). **d** The D2S/D2L ratio in the DLPFC from postmortem human brains of healthy subjects was similar across genotypes ($n = 199$; one-way ANOVA, $F_{(2, 194)} = 1.06$, $p = 0.34$). Error bars represent S.E.M. *$p < 0.05$

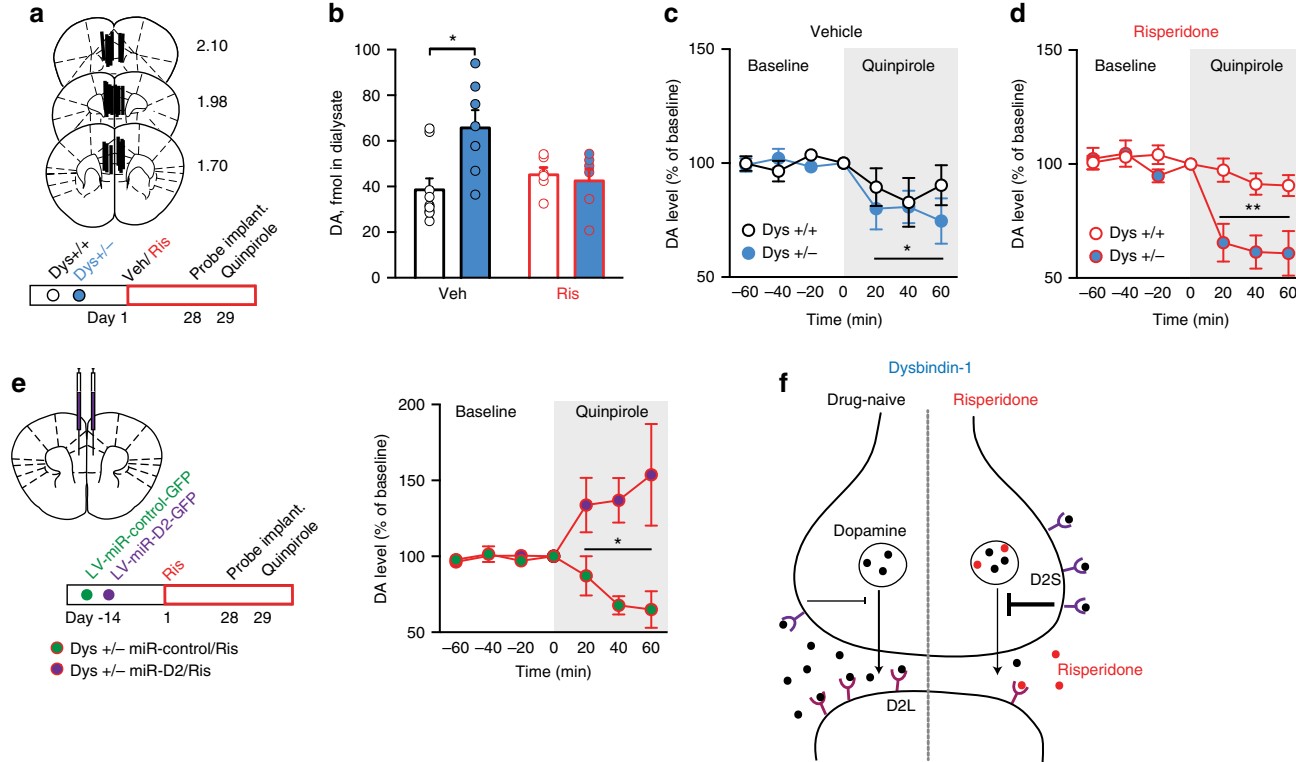

**Fig. 6** Genetic variations reducing dysbindin-1 confer unique potentiation of cortical D2 autoreceptor activity following antipsychotics. **a** Localization of probe dialyzing portion within the mPFC and timeline of the experiment. Dys+/+ and +/− mice received chronic vehicle or risperidone treatment and, after 4 weeks, were implanted with a dialysis probe for measurement of basal extracellular dopamine levels and quinpirole-induced dopamine release. **b** Increased basal extracellular dopamine levels in the mPFC of Dys+/− are restored by risperidone (two-way ANOVA, genotype × treatment, $F_{(1, 27)} = 4,61$, $p < 0.05$; $n = 6$–7/group). $p = 0.39$ vs. Dys+/+. **c** Quinpirole infusion (gray area) equally decreased extracellular dopamine release in the mPFC of Dys+/+ (white circles) and +/− (blue circles) mice after chronic treatment with Veh (two-way RM ANOVA, time, $F_{(6, 60)} = 2,83$; $p < 0.05$). *$p < 0.05$ vs. baseline. **d** Dys+/− (blue circles) mice following chronic treatment with Ris (in red) showed higher efficacy of mPFC quinpirole infusion (gray area) in reducing extracellular dopamine release (two-way RM ANOVA, time × genotype, $F_{(6, 66)} = 5.76$, $p < 0.0005$). Quinpirole had no effect on risperidone-treated +/+ mice (white circles; $p = 0.38$ vs. baseline). **$p < 0.005$ vs. Dys+/+. **e** Dys+/− mice received a synthetic microRNA (miR) to inactivate D2 receptors (LV-miR-D2, green circles) or a control miR (LV-miR-control, purple circles) in the mPFC. After 2 weeks, animals received daily injections of Veh or Ris for 28 days (to parallel behavioral experiment in Figs. 2, 4) and then have been implanted with a dialysis probe. On the following day in vivo microdialysis quinpirole-induced dopamine release was measured. In Dys+/− treated with Ris, D2 silencing even increased dopamine release. Injection of a control miR in Dys+/− Ris-treated mice further confirmed a decrease of quinpirole-mediated dopamine release (two-way RM ANOVA, time × group effect, $F_{(12, 78)} = 4.05$, $p < 0.0005$; $n = 5$–7 each group). Error bars represent S.E.M. *$p < 0.05$, **$p < 0.005$. **f** Figure model. In basal drug-naive conditions, genetic variations resulting in reduced dysbindin-1 expression increases tonic extracellular dopamine levels (black dots). Long-term administration of risperidone (red dots) alters the balance between short and long isoforms of D2 receptors, resulting in the potentiation of D2 presynaptic signaling. Antipsychotic drugs accumulate in synaptic vesicles and are secreted from upon exocytosis[49]. Moreover, antipsychotic drugs preferentially bind D2L postsynaptic receptors[47, 63], which might cause D2L/D2S imbalance in favor of D2S autoreceptors[13]

an imbalance of the D2 isoforms towards elevated expression of D2S in the PFC of both D2L+/− mutant and Dys+/− × D2L+/− double mutant mice (Fig. 7b). We then tested the executive functions of wild-type, Dys+/−, D2L+/−, and double Dys+/−D2L+/− littermates in the ID/ED *Operon* task. Double Dys+/−D2L+/− mutant mice outperformed wild-type controls and Dys+/− animals exclusively in the EDS stage (Fig. 7c, d). By contrast, D2L+/− single mutants showed the same EDS performance as wild-type littermates (Fig. 7c, d). These findings demonstrate that the imbalanced D2S/D2L ratio produced by chronic treatment with antipsychotics in mice with a genetic reduction of Dys leads to improvements in attentional set shifting.

## Discussion

The main finding of this study is that genetic variations associated with reduced Dys expression provide a background for a more

favorable cognitive executive functions response to antipsychotic drugs. We show that this pharmacogenetic interaction mechanistically relies on an enhancement of presynaptic cortical dopamine/D2 signaling through an imbalance of the D2S and D2L isoforms.

The NIMH reports that the cost of treating patients with schizophrenia in the US is nearly $19 billion a year (one-fourth of all mental health costs). On the European continent, the annual mean cost per patient is estimated to range from approximately €7000 to €40,000, depending on the country[53]. Antipsychotics are the frontline drugs for the management of schizophrenia and are effective in treating acute and chronic symptoms of schizophrenia, as well as reducing the risk of psychotic relapses, suicidal behavior, and hospitalization[53,54]. However, the percentage of patients able to integrate in the community and experience stable remission is only 30%, especially due to cognitive deficits, which are especially resistant to treatment[55]. Indeed, the effect of antipsychotic medications on cognitive function is less clear, despite

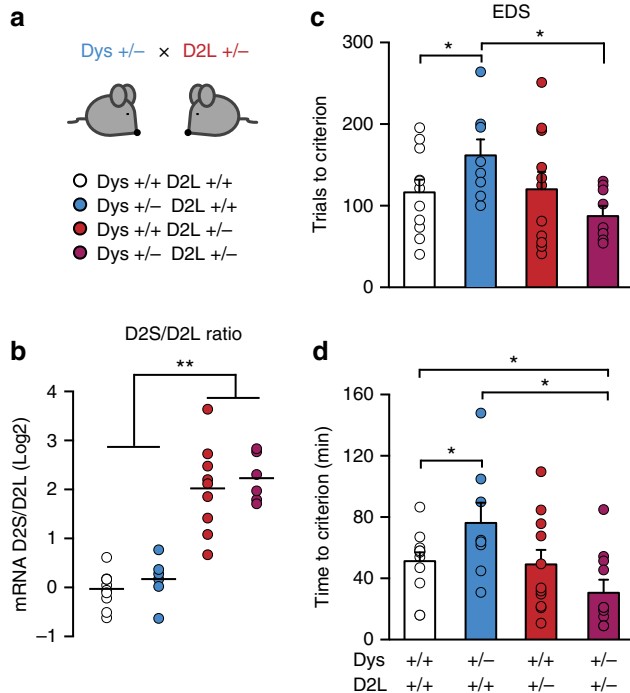

**Fig. 7** Altering the D2S/D2L ratio in favor of D2S improves executive functions exclusively in mice with genetic reduction of dysbindin-1. **a** Dys +/−D2L+/− double mutant mice (magenta bar), single Dys+/− (blue bar), single D2L+/− (red bar) and wild-type (white bar) littermates were generated by mating double heterozygous males with naive C57 female mice. **b** D2S/D2L ratio was increased in the mPFC of D2L single mutant (Dys+/+D2L+/−) and Dys+/−D2L+/− double mutant mice compared to wild-type controls (n = 6–10 each group; one-way ANOVA, $F_{(3, 29)}$ = 10.44, p < 0.0005). **c** Dys+/-D2L+/- double mutant mice needed less trials to solve the EDS compared to Dys+/− mice (unpaired t-test, t = 3.13, df = 13, p < 0.05 n = 8–12 each group. One-way ANOVA, $F_{(3, 33)}$ = 2.30, p = 0.09). **d** Dys+/-D2L+/- double mutant mice completed the EDS in less time compared to Dys+/− single mutant mice and wild-type controls (one-way ANOVA, $F_{(3, 36)}$ = 3.64, p < 0.05; n = 8–12 each group). Error bars represent S.E.M. *p < 0.05, **p < 0.005

the fact that cognitive impairments in schizophrenia are strong predictors of poor function and outcome[5,53]. Moreover, the capacity for self-care is related to cognitive performance, and once hospitalization has occurred, the presence of cognitive impairment slows the rate of the overall clinical improvement, lengthening the hospital stay[56]. Considering the overall highly heterogeneous population of patients with schizophrenia, it has been reported that treatment with first-generation and second-generation antipsychotics produces small but consistent neurocognitive improvements[6,7]. Further investigations have highlighted that cognitive functions in patients with chronic schizophrenia show marked heterogeneity[11,12], spanning from normal to impaired. Many genetic and environmental factors likely underlie these variable responses to schizophrenia treatments. Our current findings highlight one of the mechanisms that could be used to identify a subset of patients with schizophrenia whose executive functions are likely to respond better to antipsychotics, based on Dys mechanisms. This could affect a significant number of patients, as the Dys haplotype is a common genotype (Supplementary Table 2). Ultimately, these results might be potentially applied to increase the effectiveness of antipsychotics and reduce the duration of the empirical testing often required to select appropriate medication or doses of antipsychotic drugs. Our findings are in keeping with the idea of improving the use of existing drugs[57], with the final goal of implementing a precision medicine approach[37]. This study supports the idea that focusing on the genetics of individual patients rather than on the overall population is applicable to psychiatric disorders, analogous to what has been achieved with chemotherapeutic strategies in cancer genomics.

Our findings complement previous evidence that points to genetic variations influencing pharmacokinetic factors as a source of variability in the responses to antipsychotic drugs. In particular, risperidone is transformed primarily by hepatic metabolism via the cytochrome drug-metabolizing gene CYP2D6[58] to its major active metabolite, 9-hydroxyrisperidone or paliperidone. The CYP2D6 gene is highly polymorphic, resulting in large individual differences in CYP2D6 enzymatic activity[59,60]. Similarly, genetic variants in CYP3A, CYP1A2, and the ABCB1 membrane transporter, P-glycoprotein, have also been suggested to regulate the bioavailability of antipsychotic drugs[61]. Moreover, a recent study highlighted an interaction between pharmacogenetic factors and antipsychotic influences on potassium channel activity as another potential source of individual variation in response[62]. A complete assessment of genetically determined pharmacodynamic and pharmacokinetic mechanisms will provide a better assessment of the variability in adverse reactions and clinical responses to antipsychotic drug treatments. Future studies should consider whether the variability in cognitive responses that we reported here might differ between the different available antipsychotic drugs, different doses used and/or the presence of other add-on drugs (e.g., anxiolytic, antidepressant, and mood-stabilizing drugs that are commonly used in combination with antipsychotic drugs). Finally, the newer antipsychotic drugs such as clozapine target many other neurotransmitter systems that might also contribute to response variability.

Our data suggest a genetic parsing of variability in cognitive response to antipsychotic drugs that converges on the basic pharmacology of these medications. These effects might not be related to the genetic causes of schizophrenia, as revealed by current GWAS. This mirrors findings in oncology and cardiovascular medicine[36] in which genetic variations predicting therapeutic responses are not always linked to the causes of the pathology. In particular, we identified in the increased relative cortical expression of the D2S isoform a possible mechanism to explain the better executive-function responses to antipsychotic drugs in subjects with Dys reduction. Therapeutic responses to antipsychotic drugs are highly dependent on a strict range of D2 occupancy, with lower or higher binding being potentially detrimental to neurocognitive functions[13–15]. Moreover, it has been suggested that antipsychotic drugs preferentially bind D2L receptors[47,63]. Thus, chronic preferential antagonism of D2L receptors might shift the D2L/D2S balance toward the D2S isoform. The elevated propensity toward this process brought by reduced Dys might be related to its impact on intracellular trafficking at the synaptic level. Indeed, antipsychotic drugs accumulate in synaptic vesicles and are secreted upon exocytosis[49]. Moreover, Dys has an established role in rapid presynaptic induction of synaptic homeostasis[64]. Indeed, in vivo activity-dependent D2 functionality was enhanced in chronically treated Dys mutant mice, whereas it was dampened in wild-type animals. Finally, D2S receptors are internalized and desensitized more readily than D2L receptors[65–67]. Overall, the unique enhanced functionality of presynaptic D2 receptors in the mPFC of subjects with Dys hypofunction only after treatment with antipsychotics adds to previous evidence showing that (1) potentiation of D2 pathways in the PFC of humans, monkeys and rodents might facilitates executive functions[50–52] and that (2) the specific genetic deletion of presynaptic D2 receptors results in cognitive deficits

and deficits of LTD expression[68], essential for spatial memory consolidation, novel spatial learning, and behavioral flexibility. Thus, presynaptic D2 could be a key element to consider in the long-term effects of antipsychotic treatments and could provide insights into the failure of antipsychotics to ameliorate cognitive deficits in a large part of the population.

In conclusion, by translating our findings from humans to mice and back again, we provide evidence for a biologically supported approach to antipsychotic treatment response. This might open new avenues in the application of genetics to psychiatric disorders in order to optimize the efficacy of current available drugs and develop new ones by dissecting subpopulations of responders and non-responders based on concrete mechanistic insights.

## Methods

**Animals**. All procedures were approved by the Italian Ministry of Health (permits n. 230/2009-B and 107/2015-PR) and local Animal Use Committee and were conducted in accordance with the Guide for the Care and Use of Laboratory Animals of the National Institutes of Health and the European Community Council Directives. Three to 6-month-old male Dys heterozygous mutant mice[20] (Dys+/−) and their wild-type littermates (Dys+/+) were used. Distinct cohorts of naive mice were used for each experiment. Dys × D2L double mutant mice were generated by intercrossing Dys+/− with D2L+/− single mutant mice[47]. Mating schemes adopted the used of C57BL6J females mice with Dys+/− or Dys+/− × D2L+/− double heterozygous males for the two different colonies, respectively.

**Two-chamber operon ID/ED task**. Attentional set-shifting was tested in the two-chamber ID/ED Operon task as[42]. After random selection of the mice for the ID/ED task, all the behavioral manipulations were obtained blind to the genotype and pharmacological treatments of the animals. "Stuck-in-Set" ID/ED Paradigm. For habituation to the apparatus, in the first 2 days, mice were habituated for 45 min to the apparatus with only neutral stimuli (Habituation 1) and trained to move from one chamber to the other (Habituation 2). Any nose poke into the nose-poke holes resulted in a pellet delivery into the food receptacle. The next day, mice were trained to perform two randomly presented simple discriminations (e.g., between smooth vs. sand cardboard; light on vs. light off; peach vs. sage) so that they were familiar with the stimulus dimensions (Habituation 3). These exemplars were not used again. The mice had to reach a criterion of eight correct choices out of ten consecutive trials to complete this and each following testing stage. Performance was measured in all phases of all experiments using number of trials to reach the criterion; time (in minutes) to reach the criterion; time (in seconds) from breaking the photobeams adjacent to the automated door to a nose-poke response (latency to respond). To conduct the test, we used a stuck-in-set perseveration paradigm[42,69]. This procedure is only possible when three dimensions can be manipulated. A session started when a mouse was placed in one of the two chambers where all the stimuli were neutral. Then the transparent door was dropped to give the mouse access to the other chamber where the stimuli cues were on. The series of stages comprised a simple discrimination (SD), compound discrimination (CD), compound discrimination reversal (CDRe), IDS, IDS reversal (IDSRe), a second IDS 2 (IDS2), IDS reversal 2 (IDS2Re), EDS, and EDS reversal (EDSRe). The mice were exposed to the tasks in this order so that they could develop a set, or bias, toward discriminating between the correct and incorrect nose-poke hole.

**Drugs**. Risperidone (Sigma, Dorset, UK) was dissolved in 10 μl of acetic acid, made up to volume with physiological saline (0.9% NaCl), pH adjusted to 6 with 0.1 M NaOH and injected in a volume of 10 ml/kg of body weight. Animals were treated with risperidone or saline once daily for 14 consecutive days before the ID/ED Operon task started. Then, risperidone was dosed daily 2 h prior the behavioral task. The dose employed was selected in order to do not affect discrimination learning in wild-type rodents[70].

**Mice quantitative real-time PCR**. Mice samples were homogenized and total RNA was subsequently isolated with a PureLink RNA extraction Kit (Ambion, Life Technologies). Yield and purity was determined by absorbance at 230, 260, and 280 nm. Isolated RNA was converted to cDNA using high capacity cDNA reverse transcription kit (Life Technologies). Quantification of gene expression was performed with a Taqman Gene Expression Assay for *DNTBP1* (Mm01250289_m1, Life Technologies), *DRD2* (dopamine D2 receptor, Mm00438545_m1, Life Technologies), *DRD1* receptors (dopamine D1 receptor, Mm02620146_s1, Life Technologies), *SLC6A4* (solute carrier family 6 member 4, Mm00439391_m1, Life Technologies), for *GRIN1* (glutamate receptor, ionotropic, NMDA1, Mm00433790_m, Life Technologies), for *GRIN2A* (glutamate receptor, ionotropic, NMDA2A, Mm00433802_m1, Life Technologies), for *GRIN2B* (glutamate receptor, ionotropic, NMDA2B, Mm00433820_m1, Life Technologies), for *HTR2A* (5-hydroxytryptamine, serotonin, receptor 2A, Mm00555764_m1, Life Technologies),

as well as a Taqman endogenous control assay for *GAPDH* (Mm99999915_g1, Life Technologies), used as a housekeeping (normalizing) gene. TaqMan assay kits included optimized concentrations of primers and probes to detect the target gene expression. Quantification of gene expression of D2S and D2L forms was performed with custom made primers (*DR2 Short* Forward primer: 5′-ACGTGCCCTTCATCGTCACCCT-3′ *DR2 Short* Reverse primer: 5′-CG GGCAGCATCCTTGAGTGG-3′; *DR2 Long* Forward primer: 5′-ACGTGCCCTTCATCGTCACCC-3′; *DR2 Long* Reverse primer: 5′- TGGGTACA GTTGCCCTTGAGTGGT-3′; *GAPDH* Forward primer: 5′-AGGTCGGTGT-GAACGGATTTG-3′ Reverse primer: 5′-TGTAGACCATGTAGTTGAGGTCA-3′). Total volume reaction was 25 μl using SyBr Green Master Mix reagent (Applied Biosystems); 1–5 μl of cDNA were used as template for the reaction, with 10 μM forward and reverse primers. Both targets and *GAPDH* amplifications were performed in duplicate. Thermal cycling conditions included 40 cycles of 95 °C for 5 min, 95 °C for 10 s, 60 °C for 45 s. The levels of mRNA expression of these genes were measured by real-time quantitative RT-PCR using an ABI Prism 7900 sequence detection system with 384-well format (Applied Biosystems). Relative gene expression was quantified with the ΔΔCt Comparative method.

**Stereotaxic viral injection**. A lentiviral vector (Lv pPGK-eGFP-miR-D2, $10^8$ TU/mL, ICM—Plateforme de Vectorologie, Paris), coexpressing under the drive of the ubiquitous PGK promoter the eGFP and a miR, specifically directed against the mRNA of D2 (LV-miR-D2)[71], was used to downregulate D2 receptors in the mPFC. Mice were infused bilaterally (0.4 μL/each site, rate 0.3 μL/min) into the mPFC, on two different antero-posterior sites and two dorso-ventral sites according to the Paxinos and Franklin mouse brain atlas (AP: 1.8 and 2.0; ML: ±0.3; DV: −2.4 and −2.9 from Bregma).

**In vivo microdialysis**. Concentric dialysis probe, with a dialysis portion of 2.0 mm, were prepared with AN69 fibers (Hospal Dasco, Bologna, Italy)[72]. Mice were anesthetized with isoflurane and then placed in a stereotaxic frame (Kopf Instruments, Tujunga, CA) for the probe implantation. The probe was implanted into the mPFC, according to the Paxinos and Franklin mouse brain atlas (AP: ±1.9; ML: ±0.1; DV: −3.0 from Bregma). Microdialysis sessions started 24 h after the surgical procedures. Probes were perfused with ringer's solution (147.0 mM NaCl, 2.2 mM $CaCl_2$ and 4.0 mM KCl) at a constant flow rate of 1 μl/min. Collection of basal dialysate samples (20 μl) started 30 min after. After 60 min of basal sampling, a solution of Quinpirole 25 nM (Sigma, Dorset, UK) was administered through the probe for another hour of sampling collection. Dialysate samples (20 μl) were injected into an HPLC equipped with a reverse phase column (C8 3.5 μm, Waters, Mildford, MA, USA) and dopamine was quantified by a coulometric detector (ESA, Coulochem II, Bedford, MA). At the end of the experiment, mice were anesthetized with isoflurane and euthanized. Brains were removed and mPFC serial coronal sections were prepared with a vibratome to identify the location of the probes. All measurements were performed blind to the treatment and the genotype of the animals.

**Human subjects**. The study protocol for healthy controls and patients with schizophrenia was approved by the local ethics committee of the IRCCS Fondazione Santa Lucia of Roma. The study on patients with first episode of psychosis was approved by the Ethics Committee of the Children Hospital Bambino Gesù of Roma. All participants provided written informed assent and their parents/legal guardians, written informed consent. Three-hundred and fourteen healthy adults and 259 outpatients with schizophrenia who met DSM-IV-TR criteria (using to the Structured Clinical Interview for mental disorders, SCID-I/P) were recruited and assessed at the Santa Lucia Foundation in Rome. Forty-five first-episode psychosis patients with schizophrenia who met DSM-IV-TR criteria were recruited and assessed at the Bambino Gesù Hospital in Rome (see Supplementary Tables 1 and 2 for genotypic/demographic data). To reduce the possibility of environmental variables, only individuals born and educated in Italy and of Caucasian ethnicity were included. Healthy individuals were screened for a current or lifetime history of DSM-IV-TR Axis I and II disorders using the SCID-I/NP and SCID-II; they were also interviewed to confirm that no first-degree relatives had a history of any major psychiatric (e.g., mood disorders, schizophrenia spectrum disorders, substance abuse or dependence) or neurological disorders; and they had normal or corrected to normal vision. In patients under 18 years old, mental disorders were assessed using the schedule for affective disorders and Schizophrenia for school aged children present and lifetime version (K-SADS-PL). To implement the specificity of the first-episode psychosis diagnosis, positive, negative, disorganization, and general symptoms were assessed with the Structured Interview for Psychosis-Risk Syndromes (SIPS). Alcohol and drug use using sections J and L of the Composite International Diagnostic Interview (CIDI). Functioning was rated globally on the Childhood Global Assessment Scale (CGAS) and differentially on the Global Functioning: Social (GF:Social) and the Global Functioning: Role (GF: Role) scales. All patients were screened for autism-spectrum disorder using the autism quotient child or adolescent Baron-Cohen versions. In the case of positive screening, patients were assessed by a trained clinician on the autism diagnostic observation schedule-generic. No participants met criteria for autism-spectrum disorder. Demographic and clinical details included age, sex, age of illness onset,

illness duration, medical history including alcohol and drug use, admission and medication history. Antipsychotics dosage was converted to milligram equivalents of Chlorpromazine. The cognitive performance in the Wisconsin Card Sorting Test, a widely-used measure of prefrontal cognitive function that is sensitive to a subject's ability to generate hypotheses, establish response sets, and fluently shift sets, was separately performed by trained neuropsychologists at the IRCCS Santa Lucia Foundation or Bambino Gesù for patients with long-term schizophrenia or after first-episode of psychosis, respectively.

After the neuropsychological task, DNA was extracted and purified from whole peripheral blood using QIAamp DNA Blood Mini Kit (Quiagen). We specifically selected a DTNBP1 haplotype composed by the three-marker single nucleotide polymorphisms (SNPs) rs2619538, rs3213207, and rs1047631 or here, the 'Dys haplotype'[27]. DNA samples were submitted to genetic analysis for the SNPs on a ABI7900 real-time PCR instrument (Applied Biosystems) using Custom Taqman SNP Genotyping Assays for rs2619538 (C___3114517_10, Applied Biosystems), for rs3213207 (C__32386418_10, Applied Biosystems), rs1047631 (C___7460562_10, Applied Biosystems). For genotyping the manufacturer's suggested protocol was used. For PCR amplification and allelic discrimination, the ABI Prism 7900 HT Sequence Detection System and SDS software version 2.1 (Applied Biosystems) were used. HWE test and haplotype phase were performed using the software PLINK[73] for Windows http://pngu.mgh.harvard.edu/purcell/plink/. Heterozygous subjects for all the three markers (7% of our sample) were assigned to the Dys Hap+/− as the probability to be in this group, calculated on the haplotype frequency of the population, was higher than 65–70%. All polymorphisms were in HWH (p > 0.05; assessed by online resource http://www.tufts.edu/~mcourt01/Documents/Court%20lab%20%20HW%20calculator.xls; for genotype frequencies, see Supplementary Table 2).

**CATIE study setting and design**. The Clinical Antipsychotic Trials of Intervention Effectiveness (CATIE, ClinicalTrials.gov Identifier: NCT00014001) study was initiated by the NIMH to compare the effectiveness of antipsychotic drugs[74]. The protocol was made available to the public for comment, and a committee of scientific experts, health care administrators, and consumer advocates critiqued the study under the auspices of the NIMH. The study was conducted between January 2001 and December 2004 at 57 clinical sites in the United States. Patients were initially randomly assigned to receive olanzapine, perphenazine, quetiapine, or risperidone under double-blind conditions and followed for up to 18 months or until treatment was discontinued for any reason (phase 1). The present report is limited to phase 1 results. Eligible patients were 18 to 65 years of age; had received a diagnosis of schizophrenia, as determined on the basis of the Structured Clinical Interview of the Diagnostic and Statistical Manual of Mental Disorders (DSM-IV). Patients were excluded if they had received a diagnosis of schizoaffective disorder, mental retardation, or other cognitive disorders; had a history of serious adverse reactions to the proposed treatments; had only one schizophrenic episode; had a history of treatment resistance, defined by the persistence of severe symptoms despite adequate trials of one of the proposed treatments or prior treatment with clozapine; were pregnant or breast-feeding; or had a serious and unstable medical condition. (Detailed information about interventions are available here[75]). In this study, we included data only from patients tested from baseline to 18 months, with genotype data available.

**Human postmortem RNA sequencing and quantitative real-time PCR**. For expression of DTNBP1 we used RNAseq data from a total of 594 DLPFC from human postmortem brain tissue (for brain quality and demographic characteristics see[76]). For expression of D2L and D2S we used quantitative real-time PCR data from a total of 375 DLPFC from human postmortem brain tissue of subjects > 16 years old (for brain quality and demographic characteristics see[77]). Expression levels of mRNA were measured using an ABI Prizm 7900 sequence detection system with 384-well format (Applied Biosystems, Foster City, CA, USA) using Taqman assays- by-design and custom-made assays designed using PRIMER EXPRESS software (D2S Hs_01014210_m1, Applied Biosystems; D2L Forward primer: 3′-TGCACCGTTATCA TGAAGTCTAATG-5′, Reverse primer: 3′-CGGGCAGCCTCCACTCT-5′, Probe: 3′-AGTTTCCCAGTGAACAGG-5′). The PCR data were acquired from the Sequence Detector Software (SDS version 2.0, Applied Biosystems) and quantified by a standard curve method. Expression data were normalized to a geometric mean of three housekeeping genes[77]. The healthy controls had no known history of psychiatric illness or substance abuse or dependence and were screened for drug intoxication at time of death. Postmortem toxicology screening was performed by the medical examiner on every sample to test for illicit drug use. Prescription drug use at time of death, including antipsychotic (APD) and antidepressant medications, was also measured in post-mortem blood and/or cerebellar tissue to determine which prescribed medications were being used at the time of death. Positive results for antipsychotics were seen in 101 (out of 159) patients with schizophrenia. More details about RNA extraction, RNA-seq preprocessing, real-time PCR, expression quantification, and data processing can be found here[77,78].

**Statistics**. For animal experiments no statistical methods were used to predetermine sample sizes, although sample sizes were consistent with those from previous studies[42,79]. No explicit randomization method was used to allocate animals to experimental groups and mice were tested and data processed by an investigator blind to animal treatment identity. Statistical analyses were performed using commercial software (STATISTICA-StaSoft, 12). Results are expressed as mean ± standard error of the mean (S.E.M.) throughout. For analysis of variance in mice behavioral task we used ANOVA to examine the number of trials necessary to reach the criteria, time needed to complete each stage and the latency to respond. One-way ANOVAs for each single genotype were also performed to evaluate the variance of performance within each group through the different stages. Moreover, as in previous studies[80] planned comparisons were performed to test the source of significant interactions in the different stages. For human data, a one-way ANOVA was carried out to examine demographic variables, mRNA expression data, the number of perseverative and non-perseverative errors and the number of categories passed on the WCST across genotypes (Dys Hap −/−, +/− and +/+). Post hoc analyses were conducted using Newman–Keuls test with multiple comparisons corrections, when statistical significance emerged in the main effects or interactions. The accepted value for significance was p < 0.05. Result sheets of statistical tests from Stasoft detailing (wherever applicable) estimates of variance within each group, confidence intervals, effectiveness of pairing, comparison of variances across groups, are available upon request.

**Data availability**. The data that support the current study are available from the corresponding author upon reasonable request.

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

## Acknowledgements

We thank Drs. J.A. Assad, A. Contarino, D.M. Barch, and F. De Crescenzo for critical reading of the manuscript. We thank Dr. M. Morini, D. Cantatore, R. Navone, N. Pintori, G. Pruzzo, A. Parodi, A. Monteforte, and C. Chiabrera, for technical support. This work was supported by funding from the Istituto Italiano di Tecnologia, the Marie Curie FP7-Reintegration-Grant (FP7-268247), the Italian Ministry of Health (GR-2010-2315883), the Brain and Behavior Research Foundation (2015 NARSAD 23234), and the Compagnia di San Paolo (2015-0321).

## Author contributions

Conceptualization: D.S. and F.P.; Methodology and Investigation: D.S., R.M., M.M., S.S., R.E.S., M.A., F.M., S.G., F.P., F.Z., J.E.K., T.M.H., S.S.K., M.P., G.O., C.C., M.A.D.L., and F.P.; Resource: M.A., S.V., M.A.D.L., E.B., D.R.W., G.S., and F.P.; Writing: D.S., R.M., and F.P.; Visualization and Analysis: D.S., R.M., R.E.S., and F.P.; Supervision: D.R.W., G. S., and F.P. All the authors revised the manuscript.

## Additional information

**Competing interests:** The authors declare no competing interests.

