## [Peer Review File · Nature Communications]

Editorial Note: this manuscript has been previously reviewed at another journal that is not operating a transparent peer review scheme. This document only contains reviewer comments and rebuttal letters for versions considered at *Nature Communications*.

Reviewers' comments:

Reviewer #1 (Remarks to the Author):

In response to previous reviews, the authors eliminated the *Drosophila* findings and included additional experiments by viral or genetic manipulation of D2 receptor isoforms expression in vehicle- and risperidone-treated dysbindin deficient mice. These are steps in the right direction although I am concerned by the fact that the authors chose to address some serious issues with previous data from mice, raised by Reviewer 3, by dropping the data rather than performing the necessary controls to validate their conclusions.

Overall the revisions improve the focus of the manuscript and appear to strengthen the author's clinically-relevant argument that a D2 receptor isoform imbalance produced by antipsychotics exclusively in subjects with a genetic-driven reduction of dysbindin-1 is a possible mechanism underlying cognitive improvement in a subset of patients with schizophrenia treated with antipsychotics.

However this remains a rather complex argument, which is based on a number of assumptions that have not been sufficiently tested at the level of molecular (are other receptors affected?), cellular and synaptic resolution necessary to establish an unequivocal link between dysbindin deficiency, D2 receptor hyperactivity, synaptic plasticity and cognitive function. As such the study, while potentially interesting, is not definitive enough and it is probably better suited for a more specialized psychiatry journal. Additional data from human studies is encouraging but these studies are seriously confounded by the well-known variability of antipsychotic treatment patients receive during their lifetime (in terms of types of antipsychotic and other drugs used, doses, duration etc) that it is almost impossible to control for in the analysis. Well controlled prospective studies will be necessary to test the clinical relevance of these findings especially in face of overwhelming data that antipsychotics do not substantially improve cognition.

Reviewer #2 (Remarks to the Author):

The authors of this comprehensive and potentially important report have pursued the hypothesis that genetic variation in the dysbindin-1 gene might play a role in individual variation in response to antipsychotic drugs such as risperidone and aripiprazole--drugs that were administered to the patients studied here--and,for risperidone--to mice with variable levels of Dys expression. They conclude that level of Dys expression associated with a three SNP haplotype in humans plays a significant role in variation in response to antipsychotics, probably through a mechanism related to D2 dopamine receptors and "long (L)" or "short (S)" D2 receptor isoforms. In both the human and mouse studies described here, the antipsychotic agents have been used as "molecular probes"--although the authors go to great lengths to emphasize that they do not assume that drug response phenotypes

are necessarily related to the underlying pathophysiologic mechanism(s) involved in schizophrenia. From a pharmacologic perspective--the perspective that the editor asked that I take in my review--these authors have focused on the pharmacodynamics (PD) of the antipsychotics that they have studied, but the possibility that pharmacokinetic (PK) factors might also contribute to variation in drug effect has not been considered--or even mentioned.

1. Both risperidone and aripiprazole are substrates for metabolism catalyzed by the highly polymorphic CYP2D6 gene (both SNPs and structural variation ranging from gene deletion to multiple copies of the gene). Some of the metabolites of these drugs are active, but there have been reports of striking individual variation in plasma concentrations of total active drug and active etabolites. This source of variation in drug response should be pointed out. For risperidone, this pharmacogenetic variation can be quite striking.

2. Both drugs are also metabolized by CYP3A4/5--another drug metabolizing enzyme that can display quite striking genetic variation--particularly genetic differences between European and African populations, with Europeans often displaying loss of CYP3A5 activity. Once again, this source of variation in drug response should be mentioned.

3. Risperidone is also transported by the ABCB1 transporter (P-glycoprotein), another PK-related protein that displays functionally important genetic variation.

4. It is assumed that the human subjects studied here were not genotyped for these common pharmacogenetic polymorphisms, but that might have been helpful in addressing this possible mechanism for variation in drug response.

5. In summary, the results of this comprehensive and generally convincing series of studies performed using human subjects and mice indicates that one of the mechanisms by which antipsychotic agents act may involve genetic (haplotype-dependent) variation in dysbindin-1 expression acting through an effect on the ratio of dopamine 2 receptor short/long isoforms in the PFC. In addition to this source of variation, since the "molecular probes" used in these experiments were antipsychotic drugs, the authors should also point out the possibility of variation in the PK of these agents that is also genetically determined.

Reviewer #3 (Remarks to the Author):

In this revised manuscript submission, the authors have provided several additional data which have significantly strengthened the conclusions advanced in the manuscript. The major claim in this manuscript is that genetic variants that reduce the expression levels of Dysbindin-1 might confer an enhanced executive function response to an antipsychotic drug like risperidone is interesting. This is presumably due to the reduction in dysbindin-1 function producing an enhancement of D2 receptor presynaptic function in the frontal cortex. This observation could potentially be used as an approach to predict treatment

Reviewer #4 (Remarks to the Author):

The revised manuscript by Scheggia et al. has addressed all my previous comments and the new data have increased the robustness of the dysbindin-D2DR short/long-antipsychotic relationship with negative symptoms of schizophrenia. The potential of these findings to influence therapeutic decisions in the future is high. I do not have any concerns with the present version and I think the manuscript is ready except for some minor comments. I would like to suggest that the authors consider the following.

1) Citations in some areas are scanty and do not acknowledge the proper investigators. For example, the sentence

“ D2 receptors availability is altered by variations of the gene DTNBP1 encoding for dysbindin-1, a synaptic protein regulating synaptic vesicle and receptors recycling¹⁸⁻²⁰” does not cite the following and essential work to support part of this statement: PMIDs 21504412-25568125-25568125-21998198-23473812

Reviewers' comments:

Reviewer #1 (Remarks to the Author):

In response to previous reviews, the authors eliminated the *Drosophila* findings and included additional experiments by viral or genetic manipulation of D2 receptor isoforms expression in vehicle- and risperidone-treated dysbindin deficient mice. These are steps in the right direction although I am concerned by the fact that the authors chose to address some serious issues with previous data from mice, raised by Reviewer 3, by dropping the data rather than performing the necessary controls to validate their conclusions. Overall the revisions improve the focus of the manuscript and appear to strengthen the author's clinically-relevant argument that a D2 receptor isoform imbalance produced by antipsychotics exclusively in subjects with a genetic-driven reduction of dysbindin-1 is a possible mechanism underlying cognitive improvement in a subset of patients with schizophrenia treated with antipsychotics.

Based on the important suggestions made by previous Reviewer #3 in Nature, we completely changed our approach and data. In particular, by viral manipulations and by genetic manipulations we selectively altered the D2S/D2L balance (also based on new post mortem human brains results) to provide a more direct *in vivo* demonstration that the mechanism underlying the dysbindin-by-antipsychotics interaction rely on the hyper-functionality of presynaptic D2 receptors in the mPFC of dysbindin-mutant mice. Moreover, we used a LV-pPGK-eGFP-miR-D2 (ICM - Plateforme de Vectorologie, Paris) that has been recently validated to be selective for D2 receptors (e.g. Ingallinesi et al, Mol Psy 2014). In conclusion, this assessment could not have been achieved with the previous western blot methods which might not be as selective for D2 receptors and that cannot allow to differentiate between D2S and D2L forms. This is why we eliminated some of the old data which were not as informative as the new set of data added and that could have detracted from the principal focus of the paper.

In any case, as the previous reviewer was also asking, we addressed the validation of D2 receptor antibodies from different companies. In particular, we have evaluated the D2 Santa Cruz antibody specificity using different protein tissue lysates. We selected tissues in which qPCR highlighted low or absent expression of D2 receptor. Western blot analyses confirmed the real time results. Similar results were then obtained using another D2 antibody (Chemicon Millipore), which was used in previous publications (e.g. Ji et al PNAS 2009). For both antibodies, the predicted size of DRD2 is approximately 50 kDa, while a second band (70kDa) were found, consistent also with prior literature, mostly enriched in the membrane fraction, but not expressed in the cytosolic fraction (Wang H et al., J Comp Neurol. 2002; Liu XY et al., Neuron. 2006). These considerations might explain why the doublet was visible only in the "surface" blot. However, because we could not be 100% certain about the glycosylated/not glycosylated components, to avoid any possible misunderstanding, we could considered the quantification of only the band most commonly reported in the D2 literature.

However this remains a rather complex argument, which is based on a number of assumptions that have not been sufficiently tested at the level of molecular (are other receptors affected?), cellular and synaptic resolution necessary to establish an unequivocal link between dysbindin deficiency, D2 receptor hyperactivity, synaptic plasticity and cognitive function. As such the study, while potentially interesting, is not definitive enough and it is probably better suited for a more specialized psychiatry journal.

Regarding this critique on the “*level of molecular, cellular and synaptic investigations*”, we have additional gene expression data which show that other receptors, such as serotonergic and glutamatergic, are not affected. Based on this reviewer comment we added this info as supplementary table 4. However, we think that increasing the weight of the molecular investigation will risk further distraction for the readers, diluting the main message which is now very straight forward (as acknowledged by the other 3 reviewers).

Additional data from human studies is encouraging but these studies are seriously confounded by the well-known variability of antipsychotic treatment patients receive during their lifetime (in terms of types of antipsychotic and other drugs used, doses, duration etc) that it is almost impossible to control for in the analysis. Well controlled prospective studies will be necessary to test the clinical relevance of these findings especially in face of overwhelming data that antipsychotics do not substantially improve cognition.

We do agree with this Reviewer that human data of clinical responses to antipsychotic drugs can be variable and that the mentioned factors could affect the results. We started addressing this issue by performing a study on patients after their first episode of psychosis and who received only one single treatment for a fixed time before testing their cognitive function (Figure 2). We also checked possible effects of different dosages and different antipsychotics drugs used (Supplementary Tables 1 and 3). These relative advantages notwithstanding, we also highlighted in the revised manuscript the need of future studies specifically designed to consider other factors in the variability of antipsychotics effects including pharmacokinetics mechanisms, differences between several other antipsychotic drugs, different doses used and/or the presence of other add-on drugs (e.g. anxiolytic, antidepressive and mood stabilizers drugs that are also commonly used in combination with antipsychotic drugs).

The point which is more questionable is the statement of an absolute lack of cognitive improvements by antipsychotic drugs treatment. Indeed, It has been reported that treatments with first- and second-generation antipsychotics produce small, but consistent neurocognitive improvements, in well-controlled clinical trials in chronic and first episode schizophrenia patients (see for example: Davidson et al *Am. J. Psychiatry* 2009; Keefe et al *Arch. Gen. Psychiatry* 2007; Keefe et al *Schizophr Bull* 1999; Harvey et al *Am J Psychiatry* 2001; Riedel et al *Hum Psychopharmacol* 2010; Trampush et al *Schizophr Bull* 2015). We agree that the cognitive outcome are still not fully satisfactory and very variable. In this context, our findings might indicate a subset of patients with decreased dysbindin-1 expression which will indeed show better cognitive responses to antipsychotics drugs.

Reviewer #2 (Remarks to the Author):

The authors of this comprehensive and potentially important report have pursued the hypothesis that genetic variation in the dysbindin-1 gene might play a role in individual variation in response to antipsychotic drugs such as risperidone and aripiprazole--drugs that were administered to the patients studied here--and, for risperidone--to mice with variable levels of Dys expression. They conclude that level of Dys expression associated with a three SNP haplotype in humans plays a significant role in variation in response to antipsychotics, probably through a mechanism related to D2 dopamine receptors and "long (L)" or "short (S)" D2 receptor isoforms. In both the human and mouse studies described here, the antipsychotic agents have been used as "molecular probes"--although the authors go to great lengths to emphasize that they do not assume that drug response phenotypes are necessarily related to the underlying pathophysiologic mechanism(s) involved in schizophrenia. From a pharmacologic perspective--the perspective that the editor asked that I take in my review--these authors have focused on the pharmacodynamics (PD) of the antipsychotics that they have studied, but the possibility that pharmacokinetic (PK) factors might also contribute to variation in drug effect has not been considered--or even mentioned.

1. Both risperidone and aripiprazole are substrates for metabolism catalyzed by the highly polymorphic CYP2D6 gene (both SNPs and structural variation ranging from gene deletion to multiple copies of the gene). Some of the metabolites of these drugs are active, but there have been reports of striking individual variation in plasma concentrations of total active drug and active etabolites. This source of variation in drug response should be pointed out. For risperidone, this pharmacogenetic variation can be quite striking.
2. Both drugs are also metabolized by CYP3A4/5--another drug metabolizing enzyme that can display quite striking genetic variation--particularly genetic differences between European and African populations, with Europeans often displaying loss of CYP3A5 activity. Once again, this source of variation in drug response should be mentioned.
3. Risperidone is also transported by the ABCB1 transporter (P-glycoprotein), another PK-related protein that displays functionally important genetic variation.
4. It is assumed that the human subjects studied here were not genotyped for these common pharmacogenetic polymorphisms, but that might have been helpful in addressing this possible mechanism for variation in drug response.
5. In summary, the results of this comprehensive and generally convincing series of studies performed using human subjects and mice indicates that one of the mechanisms by which antipsychotic agents act may involve genetic (haplotype-dependent) variation in dysbindin-1 expression acting through an effect on the ratio of dopamine 2 receptor short/long isoforms in the PFC. In addition to this source of variation, since the "molecular probes" used in these experiments were antipsychotic drugs, the authors should also point out the possibility of variation in the PK of these agents that is also genetically determined.

We thank this Reviewer for her/his constructive comments. We agree that genetic variations relevant to pharmacokinetics mechanisms can also be factors in variable responses to antipsychotic drugs. This important issue has now been acknowledged in the revised introduction and discussion. Unfortunately, our patients have not been genotyped for polymorphisms of genes encoding for cytochrome enzymes. However, we highlighted through the text (introduction from line 72, discussion lines 296) that our findings add to the current literature about pharmacogenetics interactions between antipsychotics and drug-metabolizing enzymes and that an optimal approach in the development of personalized healthcare in psychiatry should consider both pharmacogenetics and pharmacodynamics mechanisms. In particular, as suggested by this reviewer, we reported a possible important implication of CYP2D6, CYP3A4/5 and ABCB1.

Reviewer #3 (Remarks to the Author):

In this revised manuscript submission, the authors have provided several additional data which have significantly strengthened the conclusions advanced in the manuscript.

The major claim in this manuscript is that genetic variants that reduce the expression levels of Dysbindin-1 might confer an enhanced executive function response to an antipsychotic drug like risperidone is interesting. This is presumably due to the reduction in dysbindin-1 function producing an enhancement of D2 receptor presynaptic function in the frontal cortex.

This observation could potentially be used as an approach to predict treatment.

We thanks this reviewer for these positive comments and the previous critiques that allowed us to strengthened and give a better focus to our study.

Reviewer #4 (Remarks to the Author):

The revised manuscript by Scheggia et al. has addressed all my previous comments and the new data have increased the robustness of the dysbindin-D2DR short/long-antipsychotic relationship with negative symptoms of schizophrenia. The potential of these findings to influence therapeutic decisions in the future is high. I do not have any concerns with the present version and I think the manuscript is ready except for some minor comments.

I would like to suggest that the authors consider the following.

1) Citations in some areas are scanty and do not acknowledge the proper investigators.

For example, the sentence

“ D2 receptors availability is altered by variations of the gene DTNBP1 encoding for dysbindin-1, a synaptic protein regulating synaptic vesicle and receptors recycling^{18–20}” does not cite the following and essential work to support part of this statement: PMIDs 21504412-25568125-25568125-21998198-23473812.

We thank Reviewer 4 for her/his positive comments and for her/his suggestions. As suggested in this second round of revision we now included the suggested references and we checked the others throughout the text. In some cases we had to choose and unfortunately to cut some because of references constraint.